# The Impact of a Video-Educational and Tele-Supporting Program on the Caregiver–Stroke Survivor Dyad During Transitional Care (D-STEPS: Dyadic Support Through Tele-Health and Educational Programs in Stroke Care): A Longitudinal Study Protocol

**DOI:** 10.3390/healthcare13162039

**Published:** 2025-08-18

**Authors:** Davide Bartoli, Francesco Petrosino, Emanuela Nuccio, Vincenzo Damico, Cristiana Rago, Mayra Veronese, Michele Virgolesi, Rosaria Alvaro, Ercole Vellone, Eleonora Lombardi, Gianluca Pucciarelli

**Affiliations:** 1Departmental Faculty of Medicine, Saint Camillus International University of Health and Medical Sciences, 00131 Rome, Italy; 2Department of Biomedicine and Prevention, University of Rome Tor Vergata, 00133 Rome, Italy; f.petrosino.75@gmail.com (F.P.); emanuela.nuccio@alice.it (E.N.); ragocristiana@gmail.com (C.R.); veronese.mayra@gmail.com (M.V.); rosaria.alvaro@uniroma2.it (R.A.); ercole.vellone@uniroma2.it (E.V.); eleonoralombardi90@gmail.com (E.L.); gianluca.pucciarelli@uniroma2.it (G.P.); 3ASST-Lecco, 23900 Lecco, Italy; v.damico87@libero.it; 4Department of Public Health, University of Naples Federico II, 80131 Naples, Italy; michele.virgolesi@libero.it; 5Faculty of Nursing and Midwifery, Wroclaw Medical University, 51-618 Wroclaw, Poland

**Keywords:** caregiver support, protocol study, stroke survivor, video-based training, transitional care, dyadic intervention, quality of life

## Abstract

**Introduction**: Stroke is a leading cause of long-term disability and substantially affects the quality of life (QoL) of both survivors and their caregivers. The transition from hospital to home is a vulnerable period characterized by discontinuity of care and insufficient caregiver support. Dyadic interventions—targeting both the survivor and caregiver—have shown promise in improving recovery outcomes. This protocol outlines a mixed-methods study to evaluate the impact of a video-based training intervention on the stroke survivor–caregiver dyad during the first year post-discharge. **Methods**: A mixed-methods design based on the TIDieR checklist will be implemented. Stroke survivors and their caregivers will be recruited from stroke units and rehabilitation hospitals across Italy prior to discharge. Approximately 150 dyads will receive a video training intervention followed by nurse-led transitional care support. Assessments will occur at baseline (T0) and at 1 (T1), 3 (T2), 6 (T3), and 12 months (T4) post-discharge. Outcomes will include physical functioning, disability, anxiety, depression, caregiver preparedness, burden, social support, sleep quality, and both generic and stroke-specific QoL. The study is supported by a grant from the Centre of Excellence for Nursing Scholarship, Rome, July 2024. **Conclusions**: Integrating caregivers into transitional care through structured training and support is essential for improving dyadic outcomes after stroke. Strengthening knowledge and preparedness in both survivors and caregivers enhances recovery, reduces caregiver burden, and may alleviate healthcare system costs associated with poor post-discharge outcomes.

## 1. Background

Stroke is the most common cause of adult disability [1] and has a significant impact on the quality of life (QOL) of survivors and caregivers [2]. Over 15 million people worldwide are affected by strokes each year [3]. According to Thrift et al. [4], over 40% of these are fatal while the other 60% of those affected are incapacitated.

Stroke has an early impact on the general health of stroke survivors [5]. It causes urinary incontinence in 25% of discharged stroke patients; bowel incontinence in 56%; post-stroke seizures in 5–6%; cognitive impairment in 10%; spasticity and hypertonicity in 60%; hemiplegic shoulder pain in 9–40%; depression in up to 70%; and problems with emotional lability and mood changes (such as anxiety, frustration, anger, and apathy) [6]. It can also result in decreased social interaction [7], which can impact negatively on QOL.

Stroke also has a large impact on the health of caregivers. Several studies have shown that stroke caregiving can be burdensome [8,9]. Following the discharge of stroke survivors from general hospitals and rehabilitation hospitals, an informal caregiver, usually a parent or friend, plays a crucial role in taking care of the stroke survivor at home [10]. Stroke caregivers often feel overloaded, unprepared to care, and lack confidence in their ability to care for their stroke survivors [11], especially during the transition period. Lower caregiver preparedness for the transition home contributes to the caregiver burden [12] and caregivers who are burdened have a lower QOL [2]. Families that are unprepared to treat stroke survivor are reported to often experience fatigue, depression, and stress due to lack of information provided by health workers in hospitals involved in the healing process [13]. These issues can cause errors in care, duplication of services, inappropriate or absent treatment for stroke survivors, and even increased risk of readmission to the hospital [14]. Several studies have demonstrated that well-prepared caregivers can significantly influence stroke survivors’ recovery and quality of life [15,16]. Healthcare providers therefore need to evaluate the preparation of informal caregivers, especially when this role is new to them.

Coleman et al. [17] define transitional care as: “a set of actions designed to ensure the co-ordination and continuity of health care as patients transfer between different locations or different levels of care”. According to a recent meta-analysis [18] on dyadic interventions of stroke survivors in transitional care, various supportive interventions (such as education on stroke and consequences, telephone support, or peer confrontation), bring short-term benefits in the first six months. To be effective, they must be individualised and lead to restoration of self-management of one’s own reality.

To determine the changes that occur on a short-term basis, it is important to implement a tailored intervention that could improve caregiver preparedness for the transition home after a stroke and consequently the outcomes of both stroke survivors and their caregivers. This suggests that a shift in rehabilitation philosophy from a patient-centred approach to a patient- and caregiver-centred approach, which empowers caregivers, may have better long-term outcomes. For this reason, specific interventions that include caregivers as well as stroke survivors are required. Stroke caregivers should be involved in the healing process during the stroke survivor’s acute treatment and after the rehabilitation admission.

D-STEPS is grounded in the theoretical framework of Dyadic Illness Management [19], which conceptualizes the patient–caregiver dyad as a single “unit of care”, and positions the nurse as a moderator who promotes dyadic balance in response to both immediate and evolving health needs. Additionally, the Comprehensive Post-Acute Stroke Services–Transitional Care (COMPASS-TC) model was adopted to structure the clinical approach for supporting caregivers and stroke survivors throughout the post-acute phase toward home reintegration. This model has demonstrated reliability over a 12-month period and has shown improved functional outcomes at 90 days post-stroke [20]. Lastly, to reinforce the role of remote support, the Nurse Navigators framework, developed in Queensland [21] and already recognized in cardiovascular care as a central element of continuity in post-stroke care [22], provided the theoretical foundation for identifying the nurse—as illustrated in our model (Figure 1)—as the most effective coordinator of dyadic transitional care within a tele-medicine-based setting.

Regarding the specific educational interventions that involve the caregivers in all post-stroke phases, it is crucial to give the caregiver the required preparedness for when stroke survivors will be discharged from rehabilitation hospital to their home. To date, several studies have implemented dyadic interventions as well as tele-support and tele-training programs during transitional care, however, the majority of these have focused on populations with heart failure, COPD, or multiple comorbidities [23,24]. As for stroke patients, to the best of our knowledge, the literature includes only studies that provided dyadic education using printed materials or face-to-face sessions prior to discharge, evaluating only qualitative outcomes [25,26], or tele-rehabilitation and tele-training interventions that assessed outcomes either solely in patients [27] or exclusively in caregivers [28]. These approaches do not align with the objective of optimizing dyadic outcomes during transitional care through tele-health interventions. In fact, as highlighted by a recent systematic review, the best post-stroke dyadic outcomes during transitional care are achieved only through combined tele-support and tele-training interventions [29]. In this context, we believe that the D-STEPS intervention may best address the dyadic needs during the transition home of stroke survivor-caregiver dyads.

### 1.1. AIMS

The purpose of this study is to: Evaluate the effect of a video-based tele-training and tele-support intervention (D-STEPS) on caregiver preparedness for post-stroke care management, as measured by the Preparedness for Caregiving Scale (PATH-25), over a 12-month follow-up period.

#### Secondary Objectives


To assess changes in the quality of life (QoL) of stroke survivors and caregivers (both generic and stroke-specific) over time, and examine the relationship between caregiver preparedness and QoL. In particular, the study will evaluate QoL trajectories within the dyad during the 12 months following hospital discharge.To evaluate the impact of the intervention on caregiver burden, anxiety, and psychological distress, using validated instruments including the Caregiver Burden Inventory and the Hospital Anxiety and Depression Scale (HADS).To investigate the effect of the intervention on additional caregiver-related outcomes, including perceived stroke recovery, social support, sleep quality, and stroke survivor outcomes such as hospitalizations, emergency service utilization, and mortality.To explore changes in stroke survivors’ functional status and disability, as measured by the Barthel Index and modified Rankin Scale (mRS).To examine the moderating role of dyadic mutuality in the relationship between caregiver preparedness and stroke survivor quality of life.To explore, through qualitative interviews, the lived experience of dyads, particularly regarding communication, adaptability, capability, and confidence during the transition from hospital to home.


### 1.2. Hypotheses

Drawing upon the dyadic illness management framework (Figure 1) and evidence from transitional care literature, the following hypotheses will be tested:H1: Stroke survivor–caregiver dyads who participate in the D-STEPS intervention will demonstrate a significant improvement in stroke-specific and generic quality of life (QoL) over a 12-month period.H2: Caregivers receiving the intervention will report significantly higher levels of preparedness (PATH-25) and mutuality (Mutuality Scale) compared to baseline.H3: The D-STEPS intervention will reduce caregiver burden (Caregiver Burden Inventory) and psychological distress (HADS) over time.H4: Stroke survivors receiving the intervention will show improved functional independence (Barthel Index) and reduced disability (mRS) over the 12-month follow-up.H5: The perceived quality of the dyadic relationship will moderate the effects of caregiver preparedness on stroke survivor QoL outcomes.H6 (Qualitative Hypothesis): We hypothesize that qualitative interviews will uncover enhanced dyadic communication, improved confidence and competence in caregiving, and an evolving sense of preparedness throughout the transition period, reflecting experiential dimensions that quantitative measures may not fully capture.

## 2. Methods

The study will be conducted using a mixed methods approach, based on the Template for Intervention Description and Replication checklist (Appendix A). It will use qualitative methods to explore the effect of video-training support on the stroke survivor-caregiver dyad during transitional care 1 month after discharge from a hospital or rehabilitation hospital. Quantitative methods will be used to explore the physical, psychological, social, quality of life, and educational function in the dyad 1 year after discharge. The study design complies with the Declaration of Helsinki and has been approved by the local Ethics Committee ‘REDACTED’.

In the quantitative approach, descriptive longitudinal observation methods will be employed, using different types of questionnaires. In the qualitative approach, the interpretation of the themes will guide the data analysis process [30]. Furthermore, Cohen’s phenomenology will also be used to explore the lived experience by the dyad in the training program and transition of care [31]. This methodology combines Husserl’s descriptive phenomenology and Gadamerian interpretive phenomenology. It was chosen as it has already been used in other studies and has the ability to facilitate a deeper understanding of both lived experiences and the meanings attributed to those experiences.

### 2.1. Sampling/Participants

Stroke survivors and their respective caregivers will be recruited before being discharged from stroke units or rehabilitation hospitals throughout Italy. We will employ consecutive sampling, inviting all stroke survivor–caregiver dyads meeting inclusion criteria as they present to participating centers. Consecutive enrollment methods have been commonly adopted in dyadic intervention trials to enhance feasibility and ecological validity in clinical settings [32,33]. Stroke survivors and caregivers will be assessed for suitability for the study based on the following inclusion and exclusion criteria.

For stroke survivors, inclusion criteria are (1) having had an ischaemic or haemorrhagic stroke (diagnosed by Computed Tomography or Magnetic Resonance Imaging) for the first time; (2) requiring rehabilitation care; (3) having had a mild or moderate stroke based on the National Institutes of Health Stroke Scale (NIHSS) [34]; (4) having consented to participate in the study. The exclusion criteria are (1) having pre-existing severe neurological deficits; (2) having pre-existing aphasia, reduced level of consciousness, or other comorbidities affecting the neuro-muscular system (e.g., Amyotrophic Lateral Sclerosis, Parkinson’s disease, multiple sclerosis, dementia, or other neuropathies); (3) suffering from cancer or other severe conditions of organ failure that could affect functional recovery and quality of life.

Regarding caregivers, inclusion criteria are (1) having been identified as the primary informal caregiver by the stroke survivors and (2) consenting to participate in the study. Caregivers will only be enrolled if the stroke survivors they will provide care for are. Those caregivers who cannot guarantee ongoing assistance will also be excluded.

It will not be possible to recruit stroke survivors who do not have caregivers or caregivers providing care for stroke survivors who do not consent to participate in the study. We define an “adequately completed intervention” as a dyad that meets the following criteria: (1) participated in the in-hospital video training session supervised by a nurse; (2) received and retained the educational materials (USB or email); (3) completed at least four of the five scheduled follow-ups (T0–T4); and (4) interacted with the transition care nurse at least twice post-discharge. This benchmark is consistent with thresholds used in similar tele-health dyadic interventions [35].

### 2.2. Intervention and Data Collection

Stroke units and rehabilitation staff (physicians, nurses, physical therapists, occupational therapists, and psychologists) will be trained through dedicated 3-h meetings held one month prior to the start of the study. These sessions will cover key concepts of transitional care, the effects of care transitions on stroke survivors and their caregivers, and strategies for implementing dyadic training. The training will be addressed to physicians and nurses working in the participating departments.

The nursing manager of each ward will identify two transition care nurses based on their professional qualifications and curriculum vitae. These nurses will receive additional written guidance from the principal investigator, including information on local financial resources available to the dyad, access to medical and psychosocial support services, and bureaucratic procedures (e.g., applications for caregiver work-related benefits). These nurses will provide in-hospital and post-discharge training and support to the dyads over a 12-month period following discharge. Although this is a single-group longitudinal study, we will minimize detection bias by ensuring that the nurse research assistants who collect follow-up data (T1–T4) are not involved in delivering the intervention. These assessors will be trained to follow standardized instructions and avoid interpretive interference, thus achieving partial blinding in line with best practices for behavioral interventions [36].

The dyadic intervention will be educational and use video-training. The intervention will begin in the hospital during admission.

### 2.3. Video Content and Delivery

The training videos will cover the following core skills: in-bed mobilization techniques; limb positioning during offloading; transfers from bed to wheelchair and vice versa; and transitions from a seated to a standing position using assistive devices. These contents are specifically designed to train the caregiver in home-based stroke care. The videos addressing early mobilization at home are intended to raise awareness and improve practical skills related to the mobilization of the stroke survivor—one of the most common and challenging consequences of the condition in daily life [37]. While mobility limitations may not be as pronounced in patients with mild to moderate stroke as depicted in the videos, the primary goal is to reduce the dyad’s sense of unpreparedness. By enhancing the caregiver’s capabilities, the intervention enables them to revisit the materials over time and formulate questions that may prompt the need for tailored nursing support.

The videos will be jointly viewed by the caregiver and the stroke survivor during the days preceding discharge, under the supervision of a nurse. This setting will allow the nurse to address any critical concerns or questions raised by the dyad during the session. The videos will be provided either via email or on a USB stick, according to the caregiver’s preference.

Each video ranges in length from 59 s to 4 min and 58 s, allowing for concise, topic-specific training. The use of brief, focused video content is supported by evidence suggesting that it enhances adherence to tele-educational interventions [38].

The nurse facilitating the video sessions will also assess the dyad’s digital health literacy, specifically their ability to use technological devices such as smartphones, computers, and tablets. In cases where the dyad demonstrates low-to-moderate digital literacy, a printed brochure will be provided and explained, detailing the steps to access videos via email or USB. This brochure will be available in both Italian and English (see Appendix A).

### 2.4. Monitoring Adherence and Nursing Support

Adherence to the intervention will be assessed through scheduled follow-ups every 30 days for 12 months, during which the dyad will be asked whether they have watched the videos, how many times, and whether they found the material useful.

Each nursing navigator will be assigned a maximum of 15 dyads, given the high workload involved in providing comprehensive social and clinical support. Different evidence shows that the optimal number of dyads in tele-interventions for nurses is 10–15 dyads per nurse.

Nursing support will be available during two daily time slots: from 8:00 to 13:30 and from 15:00 to 20:30. During these hours, dyads will be able to contact the nurse directly in case of urgent needs or book an appointment to address non-urgent concerns.

### 2.5. Scope of Nursing Navigator Support

The nursing navigator’s responsibilities will include:Scheduling post-acute follow-up visits for the stroke survivor;Adapting and modifying physiotherapy sessions according to dyadic needs;Managing bureaucratic procedures and requests for assistive devices in accordance with disability regulations;Supporting pharmacological management and nutritional care;Assisting with the dyad’s mobility within and outside the local area;Addressing relational, health, communicative, and psychological challenges (in collaboration with specialized professionals);Promoting and supporting social and family reintegration;Collaborate on a multidisciplinary level with various healthcare professionals involved in the process of comprehensively improving the quality of life of stroke survivors, such as neurologists, physiatrists, physical therapists, occupational therapists, speech therapists, and nutritionists. Interact with the dyad’s social circle, such as family members, employers, and colleagues.

Video support will be associated with documents that identify the nurse in charge of dyad support during the 12 months following hospital discharge, who will be contacted either by phone or email during those 12 months for dyad needs and clarifications. The nurse in charge of transitional care of the dyad will investigate through phone calls or emails or video calls, at timepoints T0, T1, T2, T3, and T4, the needs that emerge regarding support for daily activities, information on health services that are offered in the community to support clinical needs on the stroke survivor, and various educational needs that might emerge.

Specifically, the intervention program will consist of:T0: baseline quantitative investigation before the start of the interventionIn-hospital viewing, prior to discharge, of five videos on mobilisation techniques and movement management of the stroke survivor.Prior to discharge, clarification, and presentation by the nurse in charge of transitional care, including providing the contact information of the nurse contact person, documents, and videos.T1: first qualitative and second quantitative survey to ascertain the dyad’s needs and effective support.T2: third quantitative survey to ascertain needs and effective dyad support.T3: fourth quantitative survey to ascertain needs and effective dyad support.T4: fifth quantitative survey to ascertain the needs and effective dyad support.

The outcomes of caregivers and stroke survivors will be assessed at baseline (T0: before undergoing video viewing and nursing support during hospitalisation before discharge), and after 1 (T1), 3 (T2), 6 (T3), and 12 months (T4). The flowchart in Figure 2 shows the tracking of the data collection phase during the 12 months of intervention.

To ensure intervention fidelity across participating centers, we will adopt a multi-pronged strategy: (1) transition care nurses were selected based on their professional background and received training in transitional care protocols; (2) fidelity checklists aligned with the TIDieR framework are being used at each contact point; (3) the principal investigator will conduct random audits via phone or video call with center-based nurses to verify adherence to protocol components; and (4) all materials (videos, procedural documents) are standardized across centers. This strategy aligns with fidelity recommendations from Bellg et al. (2004) [39].

### 2.6. Baseline and Follow-Up Assessment

Sociodemographic characteristics of the stroke survivors and caregivers will be assessed at baseline. This includes age, gender, educational status, and stroke survivor-caregiver relationship, site and type of stroke, and survivors’ comorbidities. Follow-up assessment will be performed at baseline (discharge), 1, 3, 6, and 12 months after stroke survivors’ discharge from the rehabilitation hospital. Baseline and follow-up assessment will be performed by trained nurse research assistants.

### 2.7. Outcomes Measured

The primary and secondary outcomes to be assessed over the 12 months of the D-STEPS intervention are described below.

### 2.8. Primary Outcomes

Caregiver preparedness will be measured with the preparedness assessment for the transition home (PATH-25) instrument (primary outcome), developed by Camicia et al. [40]. The PATH-25 was designed to be a self-administered instrument to assess caregiver preparation/readiness before a patient’s discharge from an inpatient rehabilitation hospital. The instrument uses a 4-point Likert scale where 1 is “I have no understanding about…” and 4 “I have a lot of understanding about ….”. The sum score (1–100) or average score (1–4) of the items can be used to identify the level of caregiver preparedness, where a lower score means caregiver unpreparedness. As described by the authors, interventions should be tailored to address those items where the caregiver scores 2 or lower.

### 2.9. Secondary Outcomes

Several secondary outcomes will be evaluated with a battery of tools, all with established validity and reliability (Table 1). Specifically, we will use:

The stroke survivors’ specific QOL will be analysed using the Stroke Impact Scale 3.0 (SIS 3.0), a 59-item disease-specific instrument that measures QOL in stroke survivors [41]. All items use a 5-point Likert scale for responses and the total standardised score for each subscale ranges from 0 to 100, where a higher score means better cognitive performance. The validity and reliability of the SIS 3.0 have been established in the Italian stroke population [42]. This instrument will be used only for stroke survivors.

The WHOQOL-BRIEF [43] is a 26-item instrument that measures QOL in the following 4 dimensions: physical, psychological, social, and environmental. Examples of items are “How satisfied are you with your health?” or “How much do you enjoy life?”. Each item uses a 5-point Likert scale for responses from “very poor” or “very dissatisfied” to “very good” or “very satisfied”. Scores are transformed to a 0–100 scale, with higher scores indicating better QOL in each dimension. The WHOQOL-BRIEF has been used in stroke survivors [44] and caregivers [45]. In addition, the WHOQOL-BRIEF has been used in the Italian population, where it showed adequate validity and reliability [46]. We will administer the WHOQOL-BRIEF to both stroke survivors and caregivers.

The stroke survivors’ physical functioning will be measured using the Barthel Index (BI), which is a 10-item instrument [47] that measures physical functioning in activities of daily living (ADL). The score ranges from 0 to 100, where higher scores indicate greater independence. The Barthel index has been tested in the stroke population and showed supportive validity and reliability [48].

The degree of disability will be assessed with the modified Rankin scale (mRS). It is a 6 point disability scale [49] with possible scores ranging from 0 (None) to 5 (the patient has severe disability; bedridden, incontinent, requires continuous care). A separate category of 6 is usually added for patients who pass away. The scale has shown high degrees of validity and reliability around the world [50].

Depression and anxiety will be measured using the Hospital Anxiety and Depression Scale (HADS), which is a 14-item tool used widely to measure anxiety and depression with two specific subscales [51]. Example of items are “I still enjoy the things I used to enjoy” or “I feel cheerful”. Each item uses a four-point Likert scale for responses from “not at all” or “definitely as much” to “most of time” or “hardly at all”. Scores range from 0 to 21, with higher scores indicating a higher degree of depression. The validity and reliability of the HADS have been shown in several populations [52,53,54] and also in the Italian population [55]. Depression will be measured in both stroke survivor and caregivers.

The Caregiver Burden Inventory (CBI) is a 24-item scale that measures caregiver burden [56]. Examples of items are “Do you feel that your relative asks for more help than he/she needs?” or “Do you feel that your relative is dependent upon you?” Each item uses a 5-point Likert scale for responses, from “never” to “nearly always”. The CBI score ranges from 0 to 96, with higher scores meaning a greater burden. The CBI showed adequate validity and reliability in the Italian population [57,58].

The Multidimensional Scale of Perceived Social Support (MSPSS) consists of 12 items that are grouped into three factors: Family, Friends, and Significant Others [59]. A seven-point Likert scale with the options “strongly disagree” and “very strongly agree” us provided to measure the degree of agreement. Scores range from 1 to 7, with higher numbers indicating better felt social support, on both the total scale and each subscale [60].

Sleep quality will be assessed using the Pittsburg Sleep Quality Index [61]. The instrument consists of seven components, each of which assesses a particular clinical aspect of sleep. The scores from each component are added to give a sum score, also called a global score (range 0 to 21). Combined, these numerical scores provide the clinician with an efficient overall summary of a subject’s quality of sleep and sleep health. The instrument has shown good validity and reliability [62]. This instrument will be used in both stroke survivors and caregivers.

The Mutuality Scale (MS) [63] is a 15-item instrument that evaluates the mutuality perceived by the care receiver (stroke survivor version) and by the caregivers (caregiver version). Both the stroke survivor and caregiver versions of the MS can have a possible score of between 0 and 4, with higher scores meaning better mutuality. The MS has been tested for its psychometric properties in Italian stroke survivors and caregivers [64] and showed good reliability (between 0.89 and 0.94 in stroke survivors, and between 0.88 and 0.93 in caregivers). This instrument will be used in both stroke survivors and caregivers.

### 2.10. Qualitative Outcomes 

The lived experience of the transition of care with educational and social health support will be investigated with open-ended questions. The interview will feature field note acquisition and will be conducted at the dyad’s home one month after discharge. No time limit will be imposed in the response. Cohen’s phenomenological method [31] will be used for data analysis and interview construction. Phenomenology has been selected because it allows learning from the experiences of others. This study involves the detailed study of a topic (in our case, of the stroke survivor-caregiver dyads undergoing tele-training and tele-support in transitional care) to discover information or reach a new insight into the topic [65]. The stroke survivors and caregivers will be interviewed on the same day but at different times, as stated by different dyadic qualitative frameworks [66,67]. This is to better identify and understand overlaps and contrasts between the dyad. The phenomenological method has already been used to investigate the transition of care lived experiences of stroke survivors between care units [68]. In another study, the Cohen method was used to investigate how family caregivers experienced the donning/doffing training to enter COVID-19 intensive care units [69]. This method is the most suitable to evaluate the experience of transition of care with educational and social support. The qualitative interview will be based on a single open-ended question:

“Can you describe your experience of the transition from hospital to home with the support of the nurse and the educational videos? In which aspects of your health and couple’s life did the nursing tele-support and videos impact your experience of returning home after stroke?”

This question will be asked at one month and again at twelve months post-discharge. Responses will be compared to identify short- and long-term qualitative outcomes, illustrating how the lived experience of the dyad evolves over time as a result of the intervention.

### 2.11. Sample Size

A total sample size of 150 stroke-survivor caregiver dyads is expected. In 12-month longitudinal studies with greater than 40 items to be investigated, the dropout rate is around 40% [70].

In qualitative analysis, sampling will end when the data that emerges are saturated, as required by the Cohen method [31].

### 2.12. Statistical and Qualitative Analysis

Descriptive statistics including the mean, standard deviation, median, interquartile range, and frequencies, will be used to summarise socio-demographic and clinical data.

To address the potential for missing data due to the 12-month follow-up period, we will apply multiple imputation using chained equations (MICE), assuming missing at random (MAR). Sensitivity analyses will compare complete-case and imputed datasets. This approach is supported by Pan and Zhan (2020) [70], who highlight that attrition rates of up to 40% are common in longitudinal studies with complex interventions.

The effect before and after the intervention will be analysed using the paired student’s *t*-test. The effect of the program on different variables over time will be analysed with a linear hierarchical model. We will identify distinct trajectories of the stroke survivors’ QOL dimensions across the five time points of the study using latent class growth analysis (LCGA) with robust maximum likelihood estimation. The moderation analysis will be carried out with the Hosmer–Lemeshow test and the Z Sobel test. A *p*-value (*p*) < 0.05 will be considered to be statistically significant for all tests. Subsequently, SPSS^®^ Statistics for Windows, Version 26.0 (IBM Corp., Armonk, NY, USA), and Mplus 7.1 [71] were employed for statistical analysis.

Qualitative data will be analysed following the six processes indicated by Braun and Clarke [72]. Thematic analysis will describe the data, choose codes, and generate themes. The interpretive approach was intentionally chosen to study the data for their features and to interpret social reality through the participants’ individualised points of view within the context in which their reality is located. As specified in the methodology, the participant interviews will be performed using open-ended questions and audio will be recorded. These interviews will take place in locations of the participants’ choosing, giving them the flexibility to fully share their experiences. Researchers will participate in bracketing as part of critical reflection before conducting the interviews [31]. This procedure ensures the accuracy of the data analysis.

Researchers will also record contextual elements, participants’ non-verbal cues, and their reflections during the interviews. Sampling will continue for the phenomenological analysis [73] until data saturation is reached, suggesting redundancy of experiences. The field notes and full transcriptions of the interviews will be combined. Researchers will then undertake in-depth, individualised study of the data to identify the primary themes and any sub-themes. Following comparisons amongst researchers, participants will be shown the themes that were retrieved for confirmation.

Quantitative data will be analyzed using IBM SPSS Statistics version 29.0 for descriptive statistics, comparison tests, and mixed-effects models. Mplus version 8.9 will be employed for advanced statistical modeling, including latent variable analysis and mediation models. For the qualitative component, interviews will be transcribed verbatim and analyzed through line-by-line inductive coding, supported by NVivo version 14, to identify emerging themes and to explore the lived experience of the dyad across time.

### 2.13. Expected Results

We expect the D-STEPS intervention to:

Improve the trajectory of quality of life for both members of the dyad, particularly in the domains of physical function, emotional well-being, and perceived social support.

Increase caregiver knowledge, confidence, and preparedness for managing post-discharge care, while reducing burden and anxiety.

Strengthen dyadic mutuality and communication during the transitional period.

Foster early mobilization and autonomy in stroke survivors through accessible video-based training.

Decrease unplanned health service utilization (e.g., readmissions, emergency visits) through sustained support and monitoring.

Rich experiential data demonstrating growing confidence, adaptive strategies, and constructive communication within dyads, validating the patient-centered and relational impact of the intervention.

These outcomes align with the study’s theoretical framework and are supported by recent evidence [74,75] indicating that interventions targeting the dyad can produce synergistic benefits in chronic care transitions.

### 2.14. Ethical Consideration

The study was approved by the Ethics Committee of the ASL Campania Sud released on 03/05/2023 (Nr° Pr. 008787/2021). The study design complies with the Declaration of Helsinki. Before data collection, all study participants will receive information about the study and that their data will be confidential and will be stored in a safe place. All participants will be invited to read an informed consensus and will be assured to their confidentiality. The signed informed consent will be obtained by participants before proceeding with data collection.

Participants will be guaranteed that study results will be published in aggregated form, not allowing the identification of participants. Also, participants will be informed that they will have the right to withdraw from the study whenever they want. Research assistants will be trained to respect participants time during data collection; thus, if they see that participants become tired during data collection, they will offer a pause. A critical pathway has been identified for instances in which caregivers exhibit excessive emotional or social burden. In such cases, an occupational therapist and a psychologist will be involved to deliver a targeted intervention and assess whether the dyad is capable of continuing the program. If the outcome is negative, the data collected up to that critical point will be retained, and the reasons for study withdrawal will be documented accordingly.

## 3. Discussions

This longitudinal protocol describes a mixed method approach aims to investigate the effect of training, supported by tele-communication media at hospital discharge, on the caregiver-stroke survivor dyad for 12 months.

Discharge and rehabilitation programs are usually considered to be systems that use a patient-centred model. However, the need is increasingly felt to overcome this model and develop one that is based on the dyad-centred model, taking into account recent theories and conceptual models [19,76]. Stroke survivors and caregivers experience and navigate illness together, according to Lyons and Lee (2018) [19]. Considering the importance of the dyadic approach in stroke care [77], it is vital to fully understand the importance and effectiveness of dyadic interventions, especially those of an educational, informative, and supportive nature for the improvement of caregiver preparedness and consequently both stroke survivors’ and caregivers’ outcome.

Generally, the transition phase at home generates problems such as difficulty in managing safety and activities of daily living; and cognitive, behavioural, and emotional changes of stroke survivors [78]. For the caregiver, difficulties include confinement, loss of independence, tiredness, and inadequate time to perform caregiving tasks as well as managing the physical symptoms of the stroke survivor, for example, pain, not eating, and skin problems [78]. According to a recent literature review [79], caregivers are unprepared to taking charge of the stroke survivor as they need educational support from health professionals and on role change [80]. Several studies have found that caregivers may not be well prepared to provide appropriate care, such as monitoring symptoms, coordinating care, or recognising and intervening in case of complications [81,82,83]. Less prepared caregivers worry about care [84], feel burden, strain, and tension [85], and experience mood disturbances [86]. In addition, caregivers with less caregiving preparedness have poorer health than those with more extensive caregiving preparedness [87].

Petrizzo et al. (2023) described how dyad preparation should be started in hospital facilities as early as during intensive care, as it improves stoke survivor management and makes the dyad healthier [88]. We decided to support discharge with telephone support as this improves problem solving, QOL [89,90], and is helpful for the caregiver to cope with the caregiving process at home [78].

It is appropriate to integrate new frontiers of care that are not limited to crossing the threshold of hospital facilities. There is a need for dyad care to take place in the daily course of illness and over a long period of time, thereby reducing adverse effects on both caregivers and stroke survivors on discharge from acute care and rehabilitation hospitals.

According to a recent literature review, it has been established that simultaneous implementation of tele-rehabilitation, tele-support, and tele-training interventions significantly improves dyadic quality of life (QoL) during transitional care from hospital to home. These improvements span multiple domains, including physical, psychological, social, relational, and nutritional aspects, while also promoting appropriate use of healthcare services by reducing unnecessary hospital readmissions [29]. Moreover, these technology-based interventions appear to be beneficial in addressing the educational and support needs of the triad composed of the nursing navigator, caregiver, and stroke survivor [91]. Specifically, continuous feedback through nurse-led video coaching enhances adherence to rehabilitation treatments by promptly tailoring educational, training, and support needs of the dyad to their evolving post-stroke life [92].

Consequently, educational and support programs acquire a dynamic and adaptive character, evolving in response to the changing needs of the triad over time. Therefore, longitudinal studies are necessary to determine how personalized, time-sensitive care pathways impact specific outcomes related to dyadic quality of life.

The relevance of this intervention extends broadly. Given the increasing need to shift care to community settings through the use of technology, it represents a sustainable outcome—both ecologically and in alignment with emerging standards of care [93]—specifically tailored to the dyadic needs of stroke survivors [94,95]. These individuals require continuous, personalized, and adaptive support that evolves alongside the progression of the disease and the changing needs of the dyad, facilitated by the moderating role of the nursing navigator [91]. A sustainable, proximity-based intervention aligned with current international healthcare recommendations—which promote the decentralization of care and the empowerment of caregivers—stands as both timely and innovative [96,97].

### Strengths and Limitations

Regarding the limitations, in addition to the high attrition rate commonly observed in longitudinal designs, we may encounter self-selection bias, technological barriers typically affecting the older population characteristic of stroke, and heterogeneity among the dyads enrolled.

The strengths of the study include: a sample of approximately 150 dyads, which represents a sufficiently representative average for dyadic interventions supported by tele-medicine [98]; the long-term multidimensional assessment, which allows the intervention to be evaluated across various dimensions and offers a comprehensive perspective on the dyad as a unit; and, finally, the intervention itself, which represents a sustainable model and a highly innovative approach aligned with the future direction of global health.

## 4. Conclusions

This protocol outlines a comprehensive, technology-enhanced intervention aimed at supporting stroke survivor–caregiver dyads throughout the critical transitional care period. By integrating video-based training, tele-support, and continuous nurse-led guidance, the intervention seeks to improve caregiver preparedness and optimize rehabilitation outcomes. The use of tele-communication tools ensures continuity of care beyond hospital walls, allowing timely adaptation to evolving needs. This study embraces a dyad-centered approach, recognizing the interdependent nature of stroke recovery and caregiving. Addressing both clinical and psychosocial dimensions, the program has the potential to enhance quality of life for both members of the dyad. Moreover, it contributes to reducing preventable hospital readmissions by promoting competent and confident caregiving at home. The findings will provide evidence to support scalable and sustainable care models using digital health strategies. Ultimately, this protocol emphasizes the necessity of ongoing, personalized support that transcends traditional discharge practices. Future healthcare systems must prioritize long-term continuity and inclusivity in post-stroke care. This research represents a step forward in rethinking rehabilitation through a collaborative and technological lens.

## Figures and Tables

**Figure 1 healthcare-13-02039-f001:**
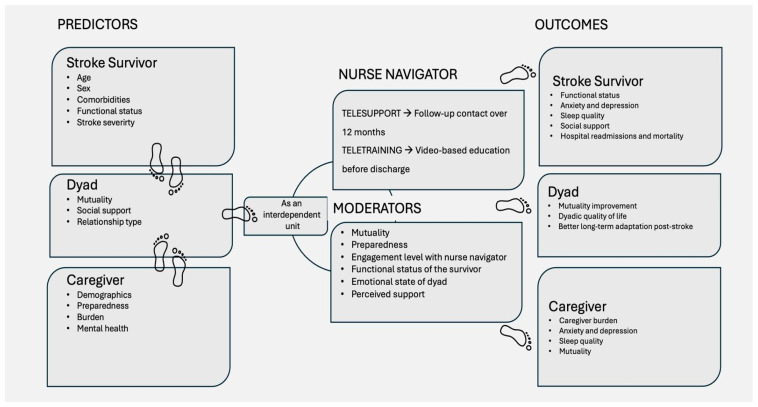
Conceptual framework of the D-STEPS protocol.

**Figure 2 healthcare-13-02039-f002:**
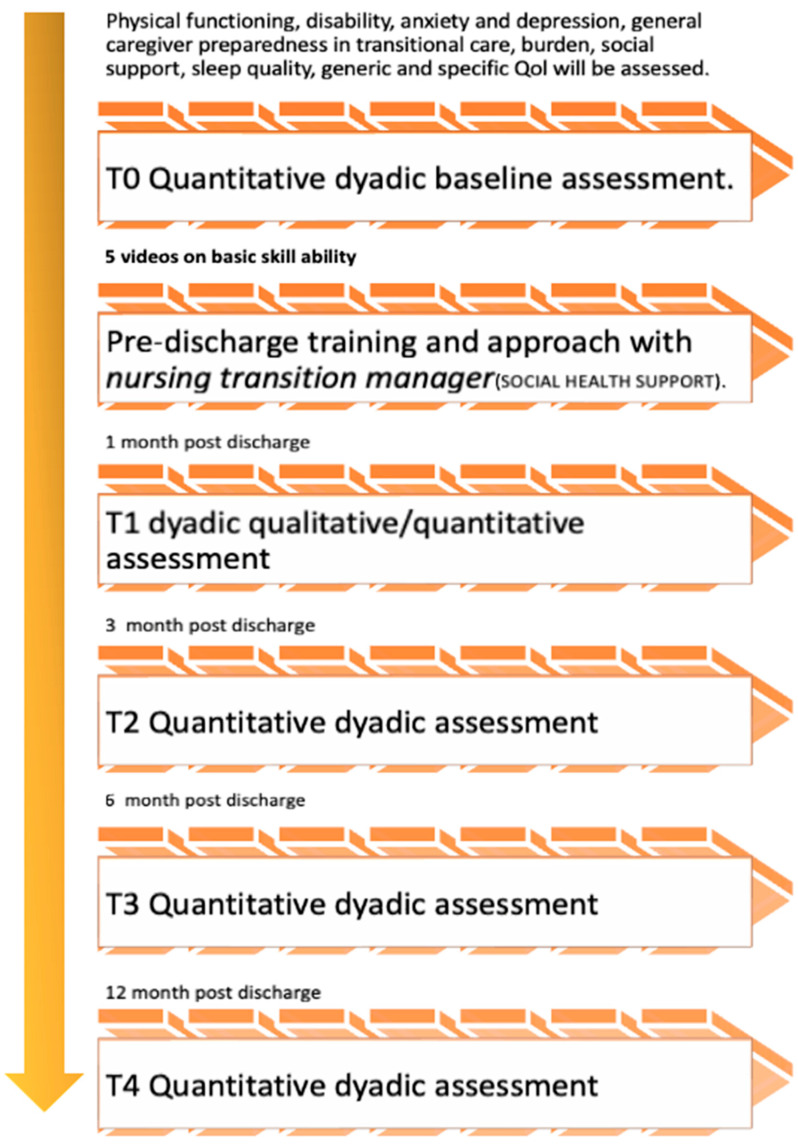
Intervention tracking and longitudinal dyadic assessments.

**Table 1 healthcare-13-02039-t001:** Variables, instruments and timing of administration.

Variables	Instruments	Time Data Collection	Administered to	
			S	C
General Characteristics				
Sociodemographic characteristics	Ad hoc Questionnaire	T0	X	X
Life condition (i.e., income, time of caregiving etc.)	Ad hoc Questionnaire	T0	X	X
Comorbities	Modified Charlson Comorbidity Index	T0	X	
Dimensions Assessed On The Dyad				
Site and type of stroke	Clinical data	T0	X	
Physical functioning	Barthel Index	T0, T1, T2, T3, T4;	X	
Disability	modified Rankin scale (mRS),	T0, T1, T2, T3, T4;	X	
Anxiety and Depression	Hospital Anxiety and Depression Scale	T0, T1, T2, T3, T4;	X	X
Caregiver preparedness in transitional care	preparedness assessment for the transition home (PATH-25)	T0, T1, T2, T3, T4;		X
Burden	Caregiver Burden Inventory	T0, T1, T2, T3,T4;		X
Social Support	Multidimensional Scale of Perceived Social Support (MSPSS),	T0, T1, T2, T3, T4;	X	X
Relationship between stroke survivors and caregivers	Mutuality scale	T0, T1, T2, T3, T4;	X	X
Sleep quality	Pittsburgh Sleep Quality Index	T0, T1, T2, T3, T4;	X	X
Generic Quality of life	WHOQOL-BREF (generic)	T0, T1, T2, T3, T4;	X	X
Specific Quality of Life	Stroke Impact Scale	T0, T1, T2, T3, T4;	X	

Note: S = stroke survivors; C = caregivers; T0 = baseline, T1 = 1 month after discharge; T2 = 3 months after discharge; T3 = 6 months after discharge and T4 = 12 months after discharge.

## Data Availability

The data that will be published may be available upon request to the corresponding author.

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
