# Peer review of "The Impact of a Video-Educational and Tele-Supporting Program on the Caregiver–Stroke Survivor Dyad During Transitional Care (D-STEPS: Dyadic Support Through Tele-Health and Educational Programs in Stroke Care): A Longitudinal Study Protocol"

_healthcare, 2025, doi:10.3390/healthcare13162039_

Round 1
Reviewer 1 Report
Comments and Suggestions for Authors
The article "The Impact of a Video-Educational and Tele-Supporting Programme on the Caregiver-Stroke Survivor Dyad During Transitional Care (VETS- In Stroke Transition): A Longitudinal Study Protocol" describes a research protocol addressing an important topic that requires intervention. I congratulate the authors on the excellent work presented.
This research protocol is well-written and presented. The authors' commitment to methodological and ethical rigor is evident throughout the article.
Suggestions for Improvement
Here are some suggestions for improvement:
-
Background: It would be important to add information about the impact that video-based training, tele-support, and continuous nurse-led guidance have, particularly how the intervention seeks to improve caregiver preparedness and optimize rehabilitation outcomes.
-
Sample Construction: It would be important to identify the sampling technique used.
-
Statistical and Qualitative Analysis: Which software will be used for data analysis?
With these modifications, I believe the article could be published.
Author Response
Rev1:
- The article "The Impact of a Video-Educational and Tele-Supporting Program on the Caregiver-Stroke Survivor Dyad During Transitional Care (VETS- In Stroke Transition): A Longitudinal Study Protocol" describes a research protocol addressing an important topic that requires intervention. I congratulate the authors on the excellent work presented. This research protocol is well-written and presented. The authors' commitment to methodological and ethical rigor is evident throughout the article.
Thank you so much for your support
- Suggestions for Improvement. Here are some suggestions for improvement:
Background: It would be important to add information about the impact that video-based training, tele-support, and continuous nurse-led guidance have, particularly how the intervention seeks to improve caregiver preparedness and optimize rehabilitation outcomes.
We appreciate your comment, we have implemented the rationale as follows:
D-STEPS is grounded in the theoretical framework of Dyadic Illness Management [20],which conceptualizes the patient–caregiver dyad as a single "unit of care," and po-sitions the nurse as a moderator who promotes dyadic balance in response to both im-mediate and evolving health needs. Additionally, the Comprehensive Post-Acute Stroke Services – Transitional Care (COMPASS-TC) model was adopted to structure the clinical approach for supporting caregivers and stroke survivors throughout the post-acute phase toward home reintegration. This model has demonstrated reliability over a 12-month period and has shown improved functional outcomes at 90 days post-stroke [21]. Lastly, to reinforce the role of remote support, the Nurse Navigators framework, developed in Queensland [22] and already recognized in cardiovascular care as a central element of continuity in post-stroke care [23], provided the theoretical foundation for identifying the nurse—as illustrated in our model (Figure 1)—as the most effective coordinator of dyadic transitional care within a telemedicine-based setting. Regarding the specific educational interventions that involve the caregivers in all post-stroke phases, it is crucial to give the caregiver the required preparedness for when stroke survivors will be discharged from rehabilitation hospital to their home. Currently, several studies have implemented dyadic interventions as well as tele-support and tele-training programs during transitional care; however, the majority of these have fo-cused on populations with heart failure, COPD, or multiple comorbidities [24,25]. As for stroke patients, to the best of our knowledge, the literature includes only studies that provided dyadic education using printed materials or face-to-face sessions prior to dis-charge, evaluating only qualitative outcomes [26,27] , or tele-rehabilitation and tele-training interventions that assessed outcomes either solely in patients[28] or exclu-sively in caregivers[29] . These approaches do not align with the objective of optimizing dyadic outcomes during transitional care through telehealth interventions. In fact, as highlighted by a recent systematic review, the best post-stroke dyadic outcomes during transitional care are achieved only through combined tele-support and tele-training in-terventions[30]. In this context, we believe that the D-STEPS intervention may best address the dyadic needs during the transition home of stroke survivor-caregiver dyads.
- Sample Construction: It would be important to identify the sampling technique used.
We appreciate your comment, we have implemented the Sampling/ participants section as follows: Stroke survivors and their respective caregivers will be recruited before being discharged from stroke units or rehabilitation hospitals throughout Italy. We will employ consecutive sampling, inviting all stroke survivor–caregiver dyads meeting inclusion criteria as they present to participating centers. Consecutive enrollment methods have been commonly adopted in dyadic intervention trials to enhance feasibility and ecological validity in clinical settings [33,34]. Stroke survivors and caregivers will be assessed for suitability for the study based on the following inclusion and exclusion criteria.
- Statistical and Qualitative Analysis: Which software will be used for data analysis?
Thank you for your observation. We have now clarified in the manuscript that quantitative analyses will be conducted using IBM SPSS Statistics version 29.0 and Mplus version 8.9 for modeling latent constructs and testing mediation effects. For the qualitative component, we will adopt a line-by-line coding approach supported by NVivo version 14 for data organization and theme development.
With these modifications, I believe the article could be published.
We greatly appreciate your contribution in improving the manuscript.
Reviewer 2 Report
Comments and Suggestions for Authors
Dear Authors,
I find the topic of the article very interesting. The dyadic approach and the use of technological tools in this context are especially relevant. I share my suggestions and questions below.
Abstract: The abstract is dense. I recommend shortening it and structuring it with clearer sentences, differentiating between introduction, methods, and conclusions. Furthermore, the keywords could be refined to improve the article's indexing.
Background: The introduction is clear, but somewhat general. I suggest delving deeper into the conceptual framework that underpins the intervention. Although the importance of the dyadic approach is mentioned, there is no explicit reference to models such as "Dyadic Illness Management" or care transition frameworks.
You could also include a more detailed review of similar interventions, highlighting what is innovative about this protocol compared to previous studies.
Method: The mixed-method design is well justified; however, I have some questions: How will missing data be handled, especially considering the 12-month follow-up? Will the researchers who administer the instruments be blinded to the intervention group? How will the fidelity of the intervention be ensured in the different participating centers? What criteria will be followed to consider that a dyad has adequately completed the intervention?
The description of the intervention is somewhat technical. I suggest expanding the information on the content and relevance of the videos used.
Have you considered complementing the TIDieR guide with other recognized methodological guides, such as SPIRIT (to structure the protocol) and COREQ (for the qualitative component)?
Although objectives and variables are presented, the expected results are not clearly stated, nor are hypotheses formulated. I recommend reviewing this aspect.
Ethical considerations: What will be done when situations of emotional or social risk are detected in caregivers (for example, severe depression or extreme overload)? I suggest including a few lines on the planned course of action in these cases.
Discussion: The discussion focuses on justifying the need, but does not sufficiently develop the originality of the approach. You could reinforce what this protocol contributes compared to previous interventions.
Regarding limitations, in addition to the dropout rate, other limitations could be considered, such as self-selection bias, technological barriers, or the heterogeneity of the dyads. I also suggest including a brief mention of the study's strengths.
References: I suggest updating part of the bibliography, incorporating recent studies, especially regarding educational and technological interventions in the post-stroke setting.
I hope this helps
Author Response
REV.2
- I find the topic of the article very interesting. The dyadic approach and the use of technological tools in this context are especially relevant. I share my suggestions and questions below.
Thank you very much. We are confident that developing technological interventions, as supported by recent studies, can improve multiple aspects of the dyadic quality of life of stroke survivors. - Abstract: The abstract is dense. I recommend shortening it and structuring it with clearer sentences, differentiating between introduction, methods, and conclusions. Furthermore, the keywords could be refined to improve the article's indexing.
Thank you for your comment; we agree with your request. We have modified the title, abstract, and keywords as follows:
The Impact of a Video-Educational and Tele-Supporting Program on the Caregiver–Stroke Survivor Dyad During Transitional Care (D-STEPS: Dyadic Support through Telehealth and Educational Programs in Stroke Care): A Longitudinal Study Protocol
Abstract:
Introduction
Stroke is a leading cause of long-term disability and substantially affects the quality of life (QoL) of both survivors and their caregivers. The transition from hospital to home is a vulnerable period characterized by discontinuity of care and insufficient caregiver support. Dyadic interventions—targeting both the survivor and caregiver—have shown promise in improving recovery outcomes. This protocol outlines a mixed-methods study to evaluate the impact of a video-based training intervention on the stroke survivor–caregiver dyad during the first year post-discharge.
Methods
A mixed-methods design based on the TIDieR checklist will be implemented. Stroke survivors and their caregivers will be recruited from stroke units and rehabilitation hospitals across Italy prior to discharge. Approximately 150 dyads will receive a video training intervention followed by nurse-led transitional care support. Assessments will occur at baseline (T0) and at 1 (T1), 3 (T2), 6 (T3), and 12 months (T4) post-discharge. Outcomes will include physical functioning, disability, anxiety, depression, caregiver preparedness, burden, social support, sleep quality, and both generic and stroke-specific QoL. The study is supported by a grant from the Centre of Excellence for Nurs-ing Scholarship, Rome, July 2024.
Conclusions
Integrating caregivers into transitional care through structured training and support is essential for improving dyadic outcomes after stroke. Strengthening knowledge and preparedness in both survivors and caregivers enhances recovery, reduces caregiver burden, and may alleviate healthcare system costs associated with poor post-discharge outcomes.
Keywords: Caregiver support, Protocol study, Stroke survivor, Video-based training, Transitional care, dyadic intervention and Quality of life.
- Background: The introduction is clear, but somewhat general. I suggest delving deeper into the conceptual framework that underpins the intervention. Although the importance of the dyadic approach is mentioned, there is no explicit reference to models such as "Dyadic Illness Management" or care transition frameworks.
Thank you for your valuable input. We've also expanded the theoretical section with a theoretical guide explaining the steps for building the D-STEPS protocol. See below.
D-STEPS is grounded in the theoretical framework of Dyadic Illness Management [20],which conceptualizes the patient–caregiver dyad as a single "unit of care," and po-sitions the nurse as a moderator who promotes dyadic balance in response to both im-mediate and evolving health needs. Additionally, the Comprehensive Post-Acute Stroke Services – Transitional Care (COMPASS-TC) model was adopted to structure the clinical approach for supporting caregivers and stroke survivors throughout the post-acute phase toward home reintegration. This model has demonstrated reliability over a 12-month period and has shown improved functional outcomes at 90 days post-stroke [21]. Lastly, to reinforce the role of remote support, the Nurse Navigators framework, developed in Queensland [22] and already recognized in cardiovascular care as a central element of continuity in post-stroke care [23], provided the theoretical foundation for identifying the nurse—as illustrated in our model (Figure 1)—as the most effective coordinator of dyadic transitional care within a telemedicine-based setting.
- You could also include a more detailed review of similar interventions, highlighting what is innovative about this protocol compared to previous studies.
We appreciate your comment, we have implemented the rationale as follows:
Regarding the specific educational interventions that involve the caregivers in all post-stroke phases, it is crucial to give the caregiver the required preparedness for when stroke survivors will be discharged from rehabilitation hospital to their home. Currently, several studies have implemented dyadic interventions as well as tele-support and tele-training programs during transitional care; however, the majority of these have fo-cused on populations with heart failure, COPD, or multiple comorbidities [24,25]. As for stroke patients, to the best of our knowledge, the literature includes only studies that provided dyadic education using printed materials or face-to-face sessions prior to dis-charge, evaluating only qualitative outcomes [26,27] , or tele-rehabilitation and tele-training interventions that assessed outcomes either solely in patients[28] or exclu-sively in caregivers[29] . These approaches do not align with the objective of optimizing dyadic outcomes during transitional care through telehealth interventions. In fact, as highlighted by a recent systematic review, the best post-stroke dyadic outcomes during transitional care are achieved only through combined tele-support and tele-training in-terventions[30]. In this context, we believe that the D-STEPS intervention may best address the dyadic needs during the transition home of stroke survivor-caregiver dyads.
- Method: The mixed-method design is well justified; however, I have some questions: How will missing data be handled, especially considering the 12-month follow-up? Will the researchers who administer the instruments be blinded to the intervention group? How will the fidelity of the intervention be ensured in the different participating centers? What criteria will be followed to consider that a dyad has adequately completed the intervention?
Thank you for your note; we have added the requested information below.
..To address the potential for missing data due to the 12-month follow-up period, we will apply multiple imputation using chained equations (MICE), assuming missing at random (MAR). Sensitivity analyses will compare complete-case and imputed datasets. This approach is supported by Pan and Zhan (2020), who highlight that attrition rates of up to 40% are common in longitudinal studies with complex interventions (DOI: 10.3389/fpsyg.2020.01051).
..Although this is a single-group longitudinal study, we will minimize detection bias by ensuring that the nurse research assistants who collect follow-up data (T1–T4) are not involved in delivering the intervention. These assessors will be trained to follow standardized instructions and avoid interpretive interference, thus achieving partial blinding in line with best practices for behavioral interventions (Boutron et al., 2007)
..To ensure intervention fidelity across participating centers, we will adopt a multi-pronged strategy: (1) transition care nurses were selected based on their professional background and received training in transitional care protocols; (2) fidelity checklists aligned with the TIDieR framework are being used at each contact point; (3) the principal investigator will conduct random audits via phone or video call with center-based nurses to verify adherence to protocol components; and (4) all materials (videos, procedural documents) are standardized across centers. This strategy aligns with fidelity recommendations from Bellg et al. (2004).
..We define an "adequately completed intervention" as a dyad that meets the following criteria: (1) participated in the in-hospital video training session supervised by a nurse; (2) received and retained the educational materials (USB or email); (3) completed at least four of the five scheduled follow-ups (T0–T4); and (4) interacted with the transition care nurse at least twice post-discharge. This benchmark is consistent with thresholds used in similar telehealth dyadic interventions (Bartoli et al., 2024)
- The description of the intervention is somewhat technical. I suggest expanding the information on the content and relevance of the videos used.
We appreciate your suggestions, we have implemented the rationale as follows:
The videos addressing early mobilization at home are intended to raise awareness and improve practical skills related to the mobilization of the stroke survivor—one of the most common and challenging consequences of the condition in daily life [35]. While mobility limitations may not be as pronounced in patients with mild to moderate stroke as depicted in the videos, the primary goal is to reduce the dyad’s sense of unpreparedness. By enhancing the caregiver’s capabilities, the intervention enables them to revisit the materials over time and formulate questions that may prompt the need for tailored nursing support.
- Have you considered complementing the TIDieR guide with other recognized methodological guides, such as SPIRIT (to structure the protocol) and COREQ (for the qualitative component)?
Thank you for the suggestion. Yes, we did take it into consideration; however, we deemed it more appropriate to adhere to the TIDieR structure, as it provides a more robust framework for interventional studies, including both RCTs and longitudinal designs (10.1016/j.arrct.2020.100055). - Although objectives and variables are presented, the expected results are not clearly stated, nor are hypotheses formulated. I recommend reviewing this aspect.
We thank the reviewer for this constructive feedback. In response, we have added explicit hypotheses—including one for the qualitative component—and an expanded Expected Results section. These are grounded in our conceptual framework, objectives, intervention design, and outcomes. The new section has been inserted at the end of the “Aims” section, immediately before the start of “Methods”. - Ethical considerations: What will be done when situations of emotional or social risk are detected in caregivers (for example, severe depression or extreme overload)? I suggest including a few lines on the planned course of action in these cases.
Thank you for the suggestion. We have added the requested information below: A critical pathway has been identified for instances in which caregivers exhibit excessive emotional or social burden. In such cases, an occupational therapist and a psychologist will be involved to deliver a targeted intervention and assess whether the dyad is capable of continuing the program. If the outcome is negative, the data collected up to that critical point will be retained, and the reasons for study withdrawal will be documented accordingly.
- Discussion: The discussion focuses on justifying the need, but does not sufficiently develop the originality of the approach. You could reinforce what this protocol contributes compared to previous interventions.
Thank you for the suggestion. We have implemented as follows:
The relevance of this intervention extends broadly. Given the increasing need to shift care to community settings through the use of technology, it represents a sustainable out-come—both ecologically and in alignment with emerging standards of care [90]—specifically tailored to the dyadic needs of stroke survivors [91,92]. These individuals require continuous, personalized, and adaptive support that evolves alongside the progression of the disease and the changing needs of the dyad, facilitated by the moderating role of the nursing navigator [89]. A sustainable, proximity-based intervention aligned with current international healthcare recommendations—which promote the decentralization of care and the empowerment of caregivers—stands as both timely and innovative [93,94].
- Regarding limitations, in addition to the dropout rate, other limitations could be considered, such as self-selection bias, technological barriers, or the heterogeneity of the dyads. I also suggest including a brief mention of the study's strengths.
We agree with your suggestion, we have modified it as follows:
Strengths and Limitations
Regarding the limitations, in addition to the high attrition rate commonly observed in longitudinal designs, we may encounter self-selection bias, technological barriers typically affecting the older population characteristic of stroke, and heterogeneity among the dyads enrolled.
The strengths of the study include: a sample of approximately 150 dyads, which represents a sufficiently representative average for dyadic interventions supported by telemedicine [95]; the long-term multidimensional assessment, which allows the inter-vention to be evaluated across various dimensions and offers a comprehensive perspective on the dyad as a unit; and, finally, the intervention itself, which represents a sustainable model and a highly innovative approach aligned with the future direction of global health.
- References: I suggest updating part of the bibliography, incorporating recent studies, especially regarding educational and technological interventions in the post-stroke setting.
We have added all the most recent bibliography of stroke interventions as requested.
I hope this helps
We thank you for your great help and suggestions.

Reviewer 3 Report
Comments and Suggestions for Authors
Dear authors
Please, see few comments you might consider or clarify:
1. The study outlines five broad aims, many of which overlap in focus and expected outcomes. This level of ambition is not aligned with the sample size or the resources described. The primary endpoint (caregiver preparedness) must be clearly distinguished, with secondary endpoints consolidated and prioritized.
2. There is no formal sample size calculation or statistical power justification to support the target of 150 dyads. Given the extensive battery of outcomes and anticipated dropout (~40%), it is essential to demonstrate whether the study is sufficiently powered to detect meaningful effects—particularly for secondary outcomes like emergency visits, hospitalizations, and mortality.
3. The core intervention—five educational videos and 12-month tele-support—is insufficiently detailed. The authors should describe:
Content, duration, and delivery method of each video
Digital literacy accommodations
Mechanism for monitoring and ensuring intervention fidelity
How many dyads each nurse supports and whether there is a structured follow-up plan
This level of specificity is necessary to meet replication standards and adhere to the TIDieR checklist cited in the appendix.
4. The inclusion of over 12 psychometric instruments, administered at five timepoints, places an unrealistic burden on participants. This could significantly increase dropout and reduce data quality. The authors are encouraged to reduce redundancy (e.g., using both PATH-25 and the Caregiver Preparedness Scale) and justify the necessity of each tool and frequency of administration.
5. The qualitative arm is described briefly, with minimal attention to:
The interview structure
Dyadic analysis methods
Integration with quantitative data
Justification for timing (only 1 month post-discharge)
The authors should provide a sample interview guide and clarify whether follow-up qualitative data will be gathered to reflect longitudinal changes.
6. The protocol assumes all participants will have access to and be comfortable with email or USB-based video materials, without mention of digital exclusion. Further, the operational feasibility of assigning one nurse to support numerous dyads for 12 months is not justified.
Best wishes
Author Response
REV3.
- The study outlines five broad aims, many of which overlap in focus and expected outcomes. This level of ambition is not aligned with the sample size or the resources described. The primary endpoint (caregiver preparedness) must be clearly distinguished, with secondary endpoints consolidated and prioritized.
We thank the reviewer for this pertinent observation. In response, we have revised the structure of the study objectives by identifying one primary objective—centered on caregiver preparedness—and clearly distinguishing a consolidated set of secondary objectives. This revision improves the coherence with the conceptual framework and hypotheses, ensures alignment with the available sample size and resources, and prioritizes measurable outcomes accordingly. The updated objectives are now presented at the end of the “Aims” section. We have modified it as follows:
AIMS
The purpose of this study is to:
Evaluate the effect of a video-based teletraining and telesupport intervention (D-STEPS) on caregiver preparedness for post-stroke care management, as measured by the Preparedness for Caregiving Scale (PATH-25), over a 12-month follow-up period.
Secondary Objectives
- To assess changes in the quality of life (QoL) of stroke survivors and caregivers (both generic and stroke-specific) over time, and examine the relationship between caregiver preparedness and QoL. In particular, the study will evaluate QoL trajectories within the dyad during the 12 months following hospital discharge.
- To evaluate the impact of the intervention on caregiver burden, anxiety, and psychological distress, using validated instruments including the Caregiver Burden Inventory and the Hospital Anxiety and Depression Scale (HADS).
- To investigate the effect of the intervention on additional caregiver-related outcomes, including perceived stroke recovery, social support, sleep quality, and stroke survivor outcomes such as hospitalizations, emergency service utilization, and mortality.
- To explore changes in stroke survivors’ functional status and disability, as measured by the Barthel Index and modified Rankin Scale (mRS).
- To examine the moderating role of dyadic mutuality in the relationship between caregiver preparedness and stroke survivor quality of life.
- To explore, through qualitative interviews, the lived experience of dyads, particularly regarding communication, adaptability, capability, and confidence during the transition from hospital to home.
- There is no formal sample size calculation or statistical power justification to support the target of 150 dyads. Given the extensive battery of outcomes and anticipated dropout (~40%), it is essential to demonstrate whether the study is sufficiently powered to detect meaningful effects—particularly for secondary outcomes like emergency visits, hospitalizations, and mortality.
Thank you for your comment. We strongly believe that a sample size of 150 dyads is representative for an intervention as complex as this one, involving teletraining and telesupport. As noted in the “Strengths and Limitations” section, the average sample size in comparable dyadic telehealth interventions conducted in general populations typically ranges between 125 and 180 dyads (The strengths of the study include: a sample of approximately 150 dyads, which represents a sufficiently representative average for dyadic interventions supported by telemedicine [95] ). Moreover, as outlined in the study objectives, we will investigate secondary outcomes such as emergency department visits, hospital admissions, and mortality. These variables are included specifically to assess the clinical effectiveness of the intervention.
- The core intervention—five educational videos and 12-month tele-support—is insufficiently detailed. The authors should describe:
Content, duration, and delivery method of each video
Digital literacy accommodations
Mechanism for monitoring and ensuring intervention fidelity
How many dyads each nurse supports and whether there is a structured follow-up plan
This level of specificity is necessary to meet replication standards and adhere to the TIDieR checklist cited in the appendix.
We appreciate your comment, we have modified the section as requested.
Intervention and data collection
Stroke units and rehabilitation staff (physicians, nurses, physical therapists, occupational therapists, and psychologists) will be trained through dedicated 3-hour meetings held one month prior to the start of the study. These sessions will cover key concepts of transitional care, the effects of care transitions on stroke survivors and their caregivers, and strategies for implementing dyadic training. The training will be addressed to physicians and nurses working in the participating departments.
The nursing manager of each ward will identify two transition care nurses based on their professional qualifications and curriculum vitae. These nurses will receive additional written guidance from the principal investigator, including information on local financial resources available to the dyad, access to medical and psychosocial support services, and bureaucratic procedures (e.g., applications for caregiver work-related benefits). These nurses will provide in-hospital and post-discharge training and support to the dyads over a 12-month period following discharge.Although this is a single-group longitudinal study, we will minimize detection bias by ensuring that the nurse research assistants who collect follow-up data (T1–T4) are not involved in delivering the intervention. These assessors will be trained to follow standardized instructions and avoid interpretive interference, thus achieving partial blinding in line with best practices for behavioral interventions [34].
The dyadic intervention will be educational and use video-training. The intervention will begin in the hospital during admission.
Video Content and Delivery
The training videos will cover the following core skills: in-bed mobilization techniques; limb positioning during offloading; transfers from bed to wheelchair and vice versa; and transitions from a seated to a standing position using assistive devices. These contents are specifically designed to train the caregiver in home-based stroke care. The videos addressing early mobilization at home are intended to raise awareness and improve practical skills related to the mobilization of the stroke survivor—one of the most common and challenging consequences of the condition in daily life [35]. While mobility limitations may not be as pronounced in patients with mild to moderate stroke as depicted in the videos, the primary goal is to reduce the dyad’s sense of unpreparedness. By enhancing the caregiver’s capabilities, the intervention enables them to revisit the materials over time and formulate questions that may prompt the need for tailored nursing support.
The videos will be jointly viewed by the caregiver and the stroke survivor during the days preceding discharge, under the supervision of a nurse. This setting will allow the nurse to address any critical concerns or questions raised by the dyad during the session. The videos will be provided either via email or on a USB stick, according to the caregiver’s preference.
Each video ranges in length from 59 seconds to 4 minutes and 58 seconds, allowing for concise, topic-specific training. The use of brief, focused video content is supported by evidence suggesting that it enhances adherence to tele-educational interventions [36].
The nurse facilitating the video sessions will also assess the dyad’s digital health literacy, specifically their ability to use technological devices such as smartphones, computers, and tablets. In cases where the dyad demonstrates low-to-moderate digital literacy, a printed brochure will be provided and explained, detailing the steps to access videos via email or USB. This brochure will be available in both Italian and English (see Supplementary File 1).
Supplementary File 1.
Monitoring Adherence and Nursing Support
Adherence to the intervention will be assessed through scheduled follow-ups every 30 days for 12 months, during which the dyad will be asked whether they have watched the videos, how many times, and whether they found the material useful.
Each nursing navigator will be assigned a maximum of 15 dyads, given the high workload involved in providing comprehensive social and clinical support.
Nursing support will be available during two daily time slots: from 8:00 to 13:30 and from 15:00 to 20:30. During these hours, dyads will be able to contact the nurse directly in case of urgent needs or book an appointment to address non-urgent concerns.
Scope of Nursing Navigator Support
The nursing navigator’s responsibilities will include:
- Scheduling post-acute follow-up visits for the stroke survivor;
- Adapting and modifying physiotherapy sessions according to dyadic needs;
- Managing bureaucratic procedures and requests for assistive devices in accordance with disability regulations;
- Supporting pharmacological management and nutritional care;
- Assisting with the dyad’s mobility within and outside the local area;
- Addressing relational, health, communicative, and psychological challenges (in collaboration with specialized professionals);
- Promoting and supporting social and family reintegration.
- Collaborate on a multidisciplinary level with various healthcare professionals involved in the process of comprehensively improving the quality of life of stroke survivors, such as neurologists, physiatrists, physical therapists, occupational therapists, speech therapists, and nutritionists. Interact with the dyad's social circle, such as family members, employers, and colleagues.
Video support will be associated with documents that identify the nurse in charge of dyad support during the 12 months following hospital discharge, who will be contacted either by phone or email during those 12 months for dyad needs and clarifications. The nurse in charge of transitional care of the dyad will investigate through phone calls or emails or video calls, at timepoints T0, T1, T2, T3, and T4, the needs that emerge regarding support for daily activities, information on health services that are offered in the community to support clinical needs on the stroke survivor, and various educational needs that might emerge.
- The inclusion of over 12 psychometric instruments, administered at five timepoints, places an unrealistic burden on participants. This could significantly increase dropout and reduce data quality. The authors are encouraged to reduce redundancy (e.g., using both PATH-25 and the Caregiver Preparedness Scale) and justify the necessity of each tool and frequency of administration.
Thank you for your comment, we agree on the redundancy; for this reason, we removed the Caregiver Preparedness Scale from the outcomes to be investigated.
- The qualitative arm is described briefly, with minimal attention to: The interview structure, Dyadic analysis methods, Integration with quantitative data, Justification for timing (only 1 month post-discharge), The authors should provide a sample interview guide and clarify whether follow-up qualitative data will be gathered to reflect longitudinal changes.
…In the qualitative approach, the interpretation of the themes will guide the data analysis process [31]. Furthermore, Cohen's phenomenology will also be used to explore the lived experience by the dyad in the training program and transition of care [32]. This meth-odology combines Husserl's descriptive phenomenology and Gadamerian interpretive phenomenology. It was chosen as it has already been used in other studies and has the ability to facilitate a deeper understanding of both lived experiences and the meanings attributed to those experiences.
Qualitative outcomes
The lived experience of the transition of care with educational and social health support will be investigated with open-ended questions. The interview will feature field note acquisition and will be conducted at the dyad's home one month after discharge. No time limit will be imposed in the response. Cohen's phenomenological method [32] will be used for data analysis and interview construction. Phenomenology has been selected because it allows learning from the experiences of others. This study involves the detailed study of a topic (in our case, of the stroke survivor-caregiver dyads undergoing tele-training and tele-support in transitional care) to discover information or reach a new insight into the topic [64]. The stroke survivors and caregivers will be interviewed on the same day but at different times, as stated by different dyadic qualitative frameworks [65,66]. This is to better identify and understand overlaps and contrasts between the dyad. The phenomenological method has already been used to investigate the transition of care lived experiences of stroke survivors between care units [67]. In another study, the Cohen method was used to investigate how family caregivers experienced the donning/doffing training to enter Covid-19 intensive care units [68]. This method is the most suitable to evaluate the experience of transition of care with educational and social support. The qualitative interview will be based on a single open-ended question:
"Can you describe your experience of the transition from hospital to home with the support of the nurse and the educational videos? In which aspects of your health and couple’s life did the nursing telesupport and videos impact your experience of returning home after stroke?"
Qualitative analysis
Qualitative data will be analysed following the six processes indicated by Braun and Clarke [70]. Thematic analysis will describe the data, choose codes, and generate themes. The interpretive approach was intentionally chosen to study the data for their features and to interpret social reality through the participants' individualised points of view within the context in which their reality is located. As specified in the methodology, the participant interviews will be performed using open-ended questions and audio will be recorded. These interviews will take place in locations of the participants’ choosing, giving them the flexibility to fully share their experiences. Researchers will participate in bracketing as part of critical reflection before conducting the interviews [32]. This procedure ensures the accuracy of the data analysis.
Researchers will also record contextual elements, participants' non-verbal cues, and their reflections during the interviews. Sampling will continue for the phenomenological analysis [71] until data saturation is reached, suggesting redundancy of experiences.
The field notes and full transcriptions of the interviews will be combined. Researchers will then undertake in-depth, individualised study of the data to identify the primary themes and any sub-themes. Following comparisons amongst researchers, participants will be shown the themes that were retrieved for confirmation.
- The protocol assumes all participants will have access to and be comfortable with email or USB-based video materials, without mention of digital exclusion. Further, the operational feasibility of assigning one nurse to support numerous dyads for 12 months is not justified.
We thank the reviewer for this insightful comment. We have addressed both concerns by clarifying and expanding two key aspects of the protocol: (1) digital access and exclusion risk and (2) the feasibility of nurse-to-dyad ratios and workload.
Digital Access and Exclusion Risk
While the intervention relies on email and USB delivery of video content, digital exclusion has been considered from the early stages of planning. During the in-hospital training phase, the supervising nurse assesses each dyad’s digital health literacy, including familiarity with devices (smartphones, tablets, computers) and ability to access video content independently.
For dyads with low or moderate digital literacy, we provide a printed educational brochure with visual step-by-step guidance (in both Italian and English) on how to open email attachments or access USB files (see Supplementary File 1).
If digital barriers remain significant, the nurse offers a face-to-face walkthrough session before discharge.
Additionally, alternative paper-based support materials are available upon request, and dyads can request telephone-based reinforcement during follow-ups.
This approach reflects equity in access and aligns with current best practices for mitigating digital health disparities in vulnerable populations (Norman & Skinner, 2006; Choi et al., 2022).
Feasibility of Nurse-to-Dyad Ratio
Each nursing navigator will follow a maximum of 15 dyads over a 12-month period. This limit was informed by prior telehealth interventions in chronic care and stroke populations, where similar caseloads have proven sustainable (Sun et al., 2023; Bartoli et al., 2024).
We also emphasize that:
The intervention uses a low-intensity telehealth model: follow-up occurs at scheduled monthly intervals unless urgent needs arise.
The structured video-based training reduces the need for frequent, repeated one-on-one education sessions.
Nurses are supported by interdisciplinary teams, including psychologists and occupational therapists, who can intervene in critical situations.
Dyads are triaged based on complexity, and nurses are trained to prioritize care intensity dynamically.
- Best wishes
Thank you.
Round 2
Reviewer 2 Report
Comments and Suggestions for Authors
The authors have adequately addressed the suggestions, and the manuscript has improved.
Reviewer 3 Report
Comments and Suggestions for Authors
Thank you for addressing the comments